# Towards Learning Implicit Symbolic Representation for Visual Reasoning

## Abstract

Visual reasoning tasks are designed to test a learning algorithm's capability to infer causal relationships, discover object interactions, and understand temporal dynamics, all from visual cues. It is commonly believed that to achieve compositional generalization on visual reasoning, an explicit abstraction of the visual scene must be constructed; for example, object detection can be applied to the visual input to produce representations that are then processed by a neural network or a neuro-symbolic framework. We demonstrate that a simple and general self-supervised approach is able to learn implicit symbolic representations with general-purpose neural networks, enabling the end-to-end learning of visual reasoning directly from raw visual inputs. Our proposed approach "compresses" each frame of a video into a small set of tokens with a transformer network. The self-supervised learning objective is to reconstruct each image based on the compressed temporal context. To minimize the reconstruction loss, the network must learn a compact representation for each image, as well as capture temporal dynamics and object permanence from temporal context. We evaluate the proposed approach on two visual reasoning benchmarks, CATER and ACRE. We observe that self-supervised pretraining is essential to achieve compositional generalization for our end-to-end trained neural network, and our proposed method achieves on par or better performance compared to recent neuro-symbolic approaches that often require additional object-level supervision.

## 1 Introduction

This paper investigates if an end-to-end trained neural network is able to solve challenging visual reasoning tasks (Zhang et al., 2021; Girdhar & Ramanan, 2019; Yi et al., 2019) that involve inferring causal relationships, discovering object relations, and capturing temporal dynamics. A prominent approach (Shamsian et al., 2020) for visual reasoning is to construct a structured and interpretable representation from the visual inputs, and then apply symbolic programs (Mao et al., 2019) or neural networks (Ding et al., 2021) to solve the reasoning task. Despite their appealing properties, such as being interpretable and easier to inject expert knowledge into the learning framework, it is practically challenging to determine what types of symbols to use and how to detect them reliably from visual data. In fact, the suitable symbolic representation for a single scene may differ significantly across different tasks: the representation for modeling a single human's kinematics (e.g. with body parts and joints) is unlikely to be the same as that for modeling group social behaviors (e.g. each pedestrian can be viewed as a whole entity). With the success of unified neural frameworks for multi-task learning (Bommasani et al., 2021), it is desirable to have a unified input interface (e.g. raw pixels) and have the neural network learn to dynamically extract suitable representations for different tasks. However, how to learn distributed representation with a deep neural network that *behaves* and *generalizes* similarly to learning methods based on symbolic representation (Zhang et al., 2021) for visual reasoning remains an open problem.

The key hypothesis we make in this paper is that a general-purpose neural network, such as Transformers (Vaswani et al., 2017), can be turned into an *implicit* symbolic concept learner with *self-supervised pre-training*. For reasoning with image and video cues, the concepts are often organized as object-centric, as objects usually serve as the basic units in visual reasoning tasks. Our proposed approach is inspired by the success of self-supervised learning of object detectors with neural networks (Burgess et al., 2019; Locatello et al., 2020; Niemeyer & Geiger, 2021) and the emergence of

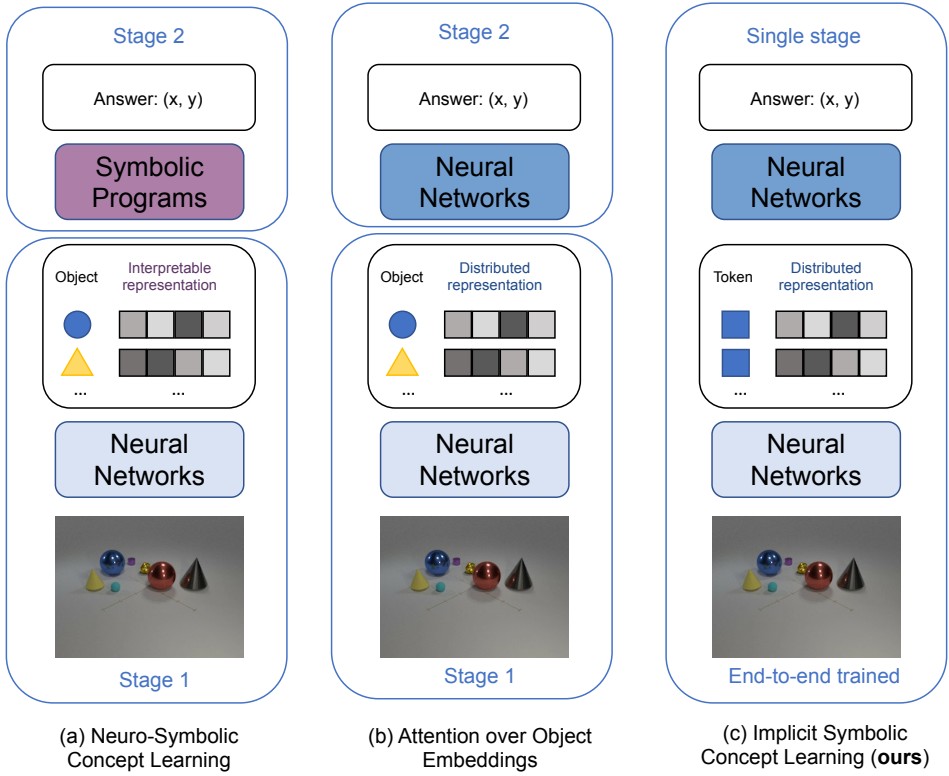

Figure 1: Comparison between a neuro-symbolic approach (e.g. Mao et al. (2019)), a hybrid approach with learned object embeddings (e.g. Ding et al. (2021)), and our proposed approach for visual reasoning. The illustration of each model family flows upwards, where visual inputs are encoded by neural networks (stage 1), and then processed by symbolic programs or another neural network to generate reasoning predictions (stage 2). Compared to (a) and (b), our approach does not require a separate "preprocessing" stage to extract the symbolic representation from visual inputs, and the self-supervised pretrained neural network can be end-to-end "finetuned" to the downstream visual reasoning tasks.

object masks in self-supervised classification networks (Caron et al., 2021). It is also motivated by *concept binding* in neuroscience (Treisman, 1996; Roskies, 1999; Feldman, 2013) and in machine learning (Greff et al., 2020), where concept binding for raw visual inputs refers to the process of *segregating* and *representing* visual scenes into a collection of (distributed) concept representation, which can be composed and utilized to solve downstream recognition and reasoning tasks. The concepts are bound in an *object-centric* fashion, where attributes (e.g. colors, shapes, sizes) of the same objects are associated via *dynamic* information routing. Different from explicit symbolic representation, implicit symbolic representation via dynamic information binding in a neural network does not require predefining the concept vocabulary or the construction of concept classifiers. The implicit representation can also be "finetuned" directly on the target tasks, it does not suffer from the early commitment or loss of information issues which may happen when visual inputs are converted into symbols and frozen descriptors (e.g. via object detection and classification).

Our proposed representation learning framework, *implicit symbolic concept learner* (IS-CL) consists of two main components: first, a single image is *compressed* into a small set of tokens with a neural network. This is achieved by a vision transformer (ViT) network (Dosovitskiy et al., 2020) with multiple "slot" tokens (e.g. the `[CLS]` token in ViT) that attend to the image inputs. Second, the slot tokens are provided as context information via a temporal transformer network for other images in the same video, where the goal is to perform video reconstruction via the *masked autoencoding* (He et al., 2022) objective with the temporal context. Despite its simplicity, the reconstruction objective motivates the emergence of two desired properties in the pretrained network: first, to provide context useful for video reconstruction, the image encoder must learn a compact representation of the scene with its slot tokens. Second, to utilize the context cues, the temporal

transformer must learn to associate objects and their implicit representation across time ("implicit tracking"), and also capture the notion of object permanence – the existence of an object even when it is occluded from the visual observations. One intuitive way to view our proposed IS-CL framework is from the perspective of Slot Attention model by Locatello et al. (2020): Instead of using a shared slot attention module to iteratively refine the encoded tokens, our image encoder is implemented as a stack of Transformer encoder layers with dedicated "slot" tokens. This generalization enables us to directly transfer the pretrained implicit symbolic representation encoded by expressive ViT backbones directly to downstream reasoning tasks.

To validate our proposed framework, we conduct extensive ablation experiments on the Compositional Actions and TEmporal Reasoning (CATER) (Girdhar & Ramanan, 2019) benchmark and the Abstract Causal REasoning (ACRE) (Zhang et al., 2021) benchmark. We observe that the self-supervised representation learned by IS-CL indeed behave likes the symbolic representation, in the sense that when finetuned on CATER and ACRE, our learned representation achieves competitive or better generalization performance when compared with the frameworks that use explicit object-centric representation. Intriguingly, we observe that the network inductive biases, such as the number of slot tokens per image, play an important role on transfer learning performance: On both datasets, we observe that a small number of slot tokens per image (1 for CATER and 4 for ACRE) lead to the best transfer learning performance on visual reasoning tasks. To the best of our knowledge, our proposed framework is the first to achieve competitive performance on CATER and ACRE without the need to construct explicit symbolic representation from visual inputs.

In summary, our paper makes the following two main contributions: First, unlike common assumptions made by neuro-symbolic approaches, we demonstrate that compositional generalization for visual reasoning can be achieved with end-to-end neural networks and implicit symbolic representations. Second, we propose a self-supervised representation learning framework IS-CL, to learn implicit symbolic representation with general-purpose Transformer neural networks. As a byproduct, we show that the learned representation achieves competitive performance on the challenging CATER and ACRE visual reasoning benchmarks. The code and pretrained checkpoints will be released upon paper acceptance.

## 2 RELATED WORK

**Neural Network Pretraining.** We have collectively made huge progress towards building unified learning frameworks for a wide range of tasks, including natural language understanding (Devlin et al., 2018; Radford et al., 2019; Brown et al., 2020; Liu et al., 2019), visual recognition (Kokkinos, 2017; Kendall et al., 2018; Zamir et al., 2018; Ghiasi et al., 2021), and multimodal perception (Jaegle et al., 2021; Sun et al., 2019; Likhosherstov et al., 2021; Girdhar et al., 2022; Alayrac et al., 2022). As this pretraining-adaptation learning paradigm gains momentum, researchers at Stanford (Bommasani et al., 2021) have even coined the term "foundation models" to refer to these pretrained neural networks. Unfortunately, most of the "foundation models" for visual data focus on perception tasks, such as object classification, detection, or image captioning.

Despite improved empirical performance on the visual question answering task (Hudson & Manning, 2019; Antol et al., 2015; Zellers et al., 2019), visual reasoning remains challenging when measured on more controlled benchmarks that require compositional generalization and causal learning (Zhang et al., 2021; Girdhar & Ramanan, 2019; Chen et al., 2022). It is commonly believed that symbolic or neurosymbolic methods (Mao et al., 2019; Yi et al., 2018; Lake & Baroni, 2018; Andreas, 2019), as opposed to the general-purpose neural networks, are required to achieve generalizable visual reasoning Yi et al. (2019); Zhang et al. (2021). To our knowledge, our proposed framework is the first to demonstrate the effectiveness of *implicit* symbolic representation on these visual reasoning benchmarks.

**Self-supervised Learning from Images and Videos.** Self-supervised learning methods aim to learn strong visual representations from unlabelled datasets using pre-text tasks. Pre-text tasks were initially hand-designed to incorporate visual priors (Doersch et al., 2015; Zhang et al., 2016; Caron et al., 2018). Subsequent works used contrastive formulations which encourage different augmented views of the same input to map to the same feature representation, whilst preventing the model from collapsing to trivial solutions (Oord et al., 2018; Chen et al., 2020; He et al., 2020; Grill et al., 2020; Akbari et al., 2021).

Our work is most related to masked self-supervised approaches. Early works in this area used stacked autoencoders (Vincent et al., 2010) or inpainting tasks (Pathak et al., 2016) with convolutional networks. These approaches have seen a resurgence recently, inspired by BERT (Devlin et al., 2018) and vision transformers (Dosovitskiy et al., 2020). BEiT (Bao et al., 2022) encodes masked patches with discrete variational autoencoders and predicts these tokens. Masked Autoencoders (MAE) (He et al., 2022), on the other hand, simply regress to the pixel values of these tokens. Masked Feature Prediction (Wei et al., 2022) (MFP) also regresses to pixelwise targets, but feature transformations of them as opposed to the direct RGB values as MAE. MAE and MFP have also both been extended to video too (Tong et al., 2022; Feichtenhofer et al., 2022), and are shown to be effective in object detection Li et al. (2022). The video reconstruction objective is also based on masked autoencoding, however, the goal is to learn a compact "implicit symbolic" representation for reasoning as opposed to generic visual descriptors for recognition tasks. We confirm empirically that the proposed method outperforms MAE and VideoMAE pretraining methods by large margins on the CATER and ACRE benchmarks.

**Object-centric Representation for Reasoning.** Most of the existing neuro-symbolic (Mao et al., 2019; Yi et al., 2018) and neural network (Ding et al., 2021) based visual reasoning frameworks require a "preprocessing" stage of symbolic representation construction, which often involves detecting and classifying objects and their attributes from image or video inputs. Our proposed framework aims to investigate the effectiveness of single-stage, end-to-end neural networks for visual reasoning, which is often more desirable than the two-stage frameworks for scenarios that require transfer learning or multi-task learning. In order to obtain the object-centric, or symbolic representation in the preprocessing stage, one can rely on a supervised object detector (Mao et al., 2019), such as Mask R-CNN (He et al., 2017). An alternative approach is to employ self-supervised objectives and learn low-level features that are correlated with objects, such as textures (Geirhos et al., 2018; Hermann et al., 2020; Olah et al., 2017), or objects themselves (Burgess et al., 2019; Locatello et al., 2020; Caron et al., 2021). In practice, supervised or self-supervised approaches for object detection and object-centric representation learning may suffer from the lack of supervised annotations, or the noisy object detection results. For example, Zhang et al. (2022) observed that object-centric representation is beneficial for transfer learning to temporal event classification only when the ground truth object detections are used.

## 3 METHOD

We now introduce the proposed implicit symbolic concept learning (IS-CL) framework. We follow the pretraining and transfer learning paradigm: During pretraining (Figure 2), we task a shared image encoder to output patch-level visual embeddings along with a small set of slot tokens that compress the image's information. The pretraining objective is masked autoencoding (MAE) for unlabeled video frames, namely reconstructing the pixel values for a subset of "masked" image patches, given the "unmasked" image patches as context. Compared to the standard MAE for images (He et al., 2022), the image decoder has access to two additional types of context information: (1) The encoded patch embedding from the unmasked image patches of the neighboring frames; (2) The encoded slot tokens from a subset of context frames. The context information is encoded and propagated by a temporal transformer network. To successfully reconstruct a masked frame, the image encoder must learn a compact representation of the full image via the slot tokens, and the temporal transformer has to learn to capture object permanence and temporal dynamics.

During transfer learning (Figure 3), the image decoder can be discarded, and only the image encoder and temporal transformer need to be transferred. The inputs to the temporal transformer are the slot tokens encoded from individual, unmasked video frames. We consider the full finetuning strategy where the weights of both the newly added task decoder (e.g. a linear classifier), and the pretrained image and temporal transformers are updated during transfer learning.

**Image Encoder:** We adopt the Vision Transformer (ViT) backbone to encode each image independently: An input image is broken into non-overlapping patches of $16 \times 16$ pixels, which are then linearly projected into patch embeddings as inputs to the transformer encoder. Spatial information is preserved by sinusoidal positional encodings. We use the standard ViT-Base configuration which has 12 Transformer encoder layers. Each layer has hidden size of 768, MLP projection size of 3072, and 12 attention heads. During pretraining, a subset of video frames are spatially masked randomly

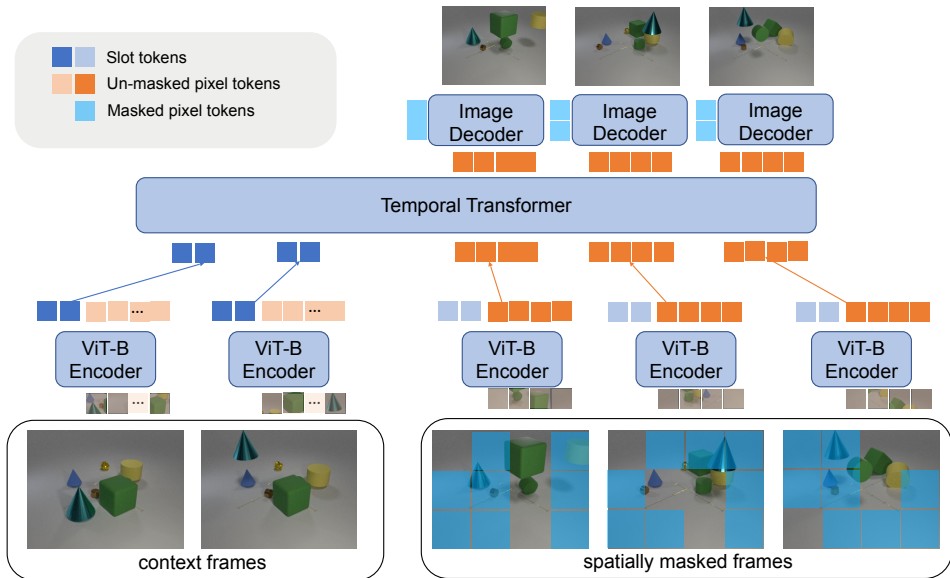

Figure 2: An overview of the proposed implicit symbolic concept learner (IS-CL) framework for self-supervised pretraining. We consider the video reconstruction objective via masked autoencoding: A ViT-B image encoder is tasked to (1) extract visual representation (orange) for the unmasked patches per image and (2) compress an image into a small set of slot tokens (blue). A temporal transformer then propagates the information from slot representation and patch-level representation from neighboring frames, which are essential for successful reconstruction. We hypothesize that implicit symbolic representation emerges automatically in the slot tokens by self-supervised pre-training.

given a masking ratio. As illustrated in Figure 2, only the unmasked image patches are fed into the ViT-B encoder. For context frames and during transfer learning, all image patches are provided as inputs to the image encoder.

**Slot Tokens:** In the seminal work by Locatello et al. (2020), slot tokens are defined as the representational bottleneck in an image autoencoder, where the slot representations are iteratively updated with a GRU after the slots attend to the visual inputs in each iteration. We borrow their terminology, and also use slots to denote the representational bottleneck which we hope to encode symbolic, or object-centric information. We generalize their slot update rules by: (1) iteratively updating the input representation from raw pixels to visual representation encoded by the Transformer encoder (ViT); (2) replacing cross-attention with multi-headed self-attention; (3) using MLP layers with untied weights to update the intermediate slot representation as opposed to a shared GRU network. These two modifications allow us to implement "slot attention" directly with a Transformer encoder, simply by prepending slot tokens as additional inputs to the encoder (similar to [CLS] tokens). The initial slot embeddings at the input of the visual encoder are implemented as a learnable embedding lookup table. To compare the effectiveness of different methods to aggregate "slot" information, we also explore single-headed soft attention and Gumbel-max attention as used by Xu et al. (2022).

**Temporal Transformer:** To propagate temporal information across frames, we use another transformer encoder (with fewer layers than the ViT-B image encoder) which takes the tokens encoded by the image encoder as its inputs. During pretraining, the slot tokens from context frames, along with the unmasked patch tokens from the query frames are concatenated together and fed into the temporal transformer. For each query image, the temporal transformer outputs its corresponding unmasked patch tokens *contextualized* from both the unmasked patches from neighboring query frames and the slot tokens from context frames. The contextualized patches are then fed into the image decoder to compute the reconstruction loss. To preserve temporal position information, we use learned positional embeddings (implemented with an embedding lookup table). During transfer learning, the temporal transformer takes the slot tokens encoded by the image encoder as its inputs.

Putting the image encoder and the temporal transformer together, the overall video encoder used for transfer learning can be viewed as an factorized space-time encoder proposed by Arnab et al. (2021). It is more parameter-efficient than the vanilla video vision transformer used by Tong et al. (2022).

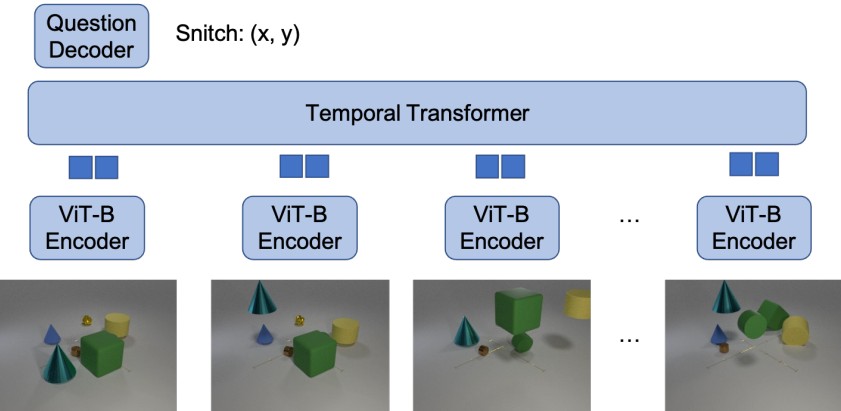

Figure 3: An illustration of the transfer learning process: Both the ViT-B image encoder and the temporal transformer are transferred to downstream visual reasoning tasks to encode video inputs. Unlike pretraining, only the slot tokens are provided as inputs to the temporal transformer.

**Image Decoder for Pre-training:** We use the same image decoder as in (He et al., 2022). As illustrated in Figure 2, the query images are decoded independently given the contextualized unmasked patch tokens. The image decoder is implemented with another transformer, where masked patch tokens are appended to the contextualized unmasked patch tokens as inputs to the image decoder. Sinusoidal positional encodings are used to indicate the spatial locations of individual patch tokens. We use the same number of layers, hidden size, and other hyperparameters as recommended by He et al. (2022). For pre-training purpose, we use mean squared error to measure the distance between the original query image patches and the reconstructed patches.

**Transfer Learning:** As the goal of pre-training is to learn the slot tokens which we hope to compress an input image into several implicitly symbolic tokens, we only ask the image encoder to generate the slot tokens during finetuning (Figure 3), which are fed to the temporal transformer as its inputs. We then average pool the output tokens of the temporal transformer and add a task-specific decoder to make predictions. Both benchmarks used in our experiments can be formulated as multi-class classification: For CATER, the goal is to predict the final location of the golden snitch (Figure 4 top), where the location is quantized into one of the 6×6 positions; For ACRE, the goal is to predict whether the platform will activate, not activate, or undetermined given a query scenario (Figure 4 bottom). We hence use linear classifiers as the task-specific decoders and the standard softmax cross-entropy for transfer learning.

## 4 EXPERIMENTS

We present results on CATER (Girdhar & Ramanan, 2019) and ACRE (Zhang et al., 2021).

### 4.1 EXPERIMENTAL SETUP

**Benchmarks:** In the classic "shell game", a ball is placed under a cup and shuffled with other empty cups on a flat surface; then, the objective is to determine which cup in the final shuffled configuration contains the ball. Inspired by this, CATER is a dataset composed of videos of CLEVR (Johnson et al., 2017) objects as they move around the scene. A special golden ball, called the "snitch", is present in each video, and the associated reasoning task is to determine the snitch's position at the final frame. Object locations in the CATER dataset are denoted by positions on an invisible 6-by-6 grid; therefore, in essence, the CATER task boils down to a 36-way classification problem. Solving this task is complicated by the fact that larger objects can visually occlude smaller ones, and certain objects can be picked up and placed down to explicitly cover other objects; when an object is covered, it changes position in consistence with the larger object that covers it. Therefore, in order to solve the task successfully, a model must learn to reason not only about objects and movement, but also about object permanence, long-term occlusions, and recursive covering relationships.

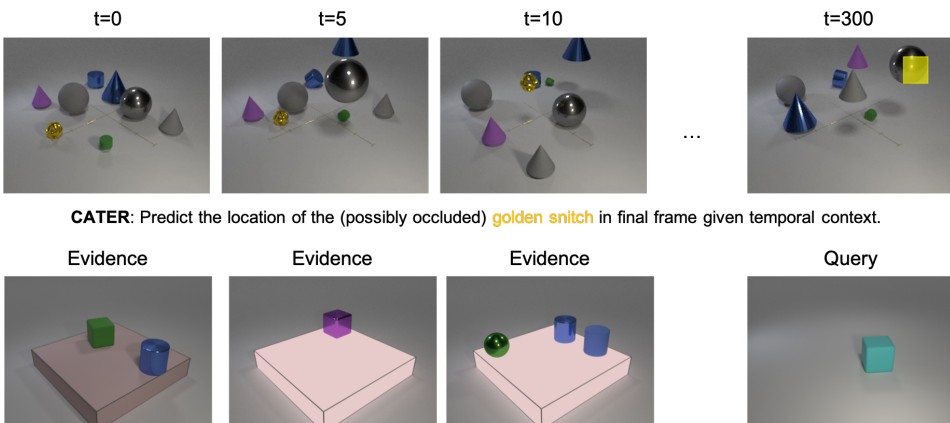

Figure 4: Illustration of the CATER and ACRE benchmarks.

The CATER dataset features a split where the camera is statically fixed to a particular angle and position throughout the videos, as well as a moving camera split where the viewing angle is able to change over time. We use the static split for evaluation. Each video has 300 frames. A visualization of a CATER video and the associated snitch localization task is shown in Figure 4 (top).

The ACRE dataset tests a model's ability to understand and discover causal relationships. The construction of the dataset is motivated by the Blicket experiment in developmental psychology (Gopnik & Sobel, 2000), where there is a platform as well as many distinct objects, some of which contain the "Blicketness" property. When at least one object with the "Blicketness" property is placed on the platform, music will be played; otherwise, the platform will maintain silence. Given a few context demonstrations of different object combinations, as well as the resulting effect, young children have been shown to successfully infer which objects contain the "Blicketness" property, and which combinations would cause the platform to play music. In ACRE, the platform is represented by a large pink block that either glows or remains dim depending on the combination of CLEVR objects placed on it. Given six evidence frames of objects placed on the platform, the objective of the reasoning task is to determine the effect a query frame, containing a potentially novel object combination, would have on the platform. Possible answers include lighting up the platform, keeping the platform dim, or unable to be determined with the given evidence frames. A visualization of an example ACRE sample is shown in the bottom row of Figure 4 (bottom).

**Pretraining data:** We use the unlabeled videos from the training and validation splits of the CATER dataset for pretraining. Both the static and moving camera splits are used, which contains 9,304 videos in total. In our experiments, we observe that ACRE requires higher resolution inputs during pretraining and finetuning. Our default preprocessing setup is to randomly sample 32 frames of 64×64 for pretraining checkpoints to be transferred to CATER, and 16 frames of 224×224 for pretraining checkpoints to be transferred to ACRE. The randomly sampled frames are sorted to preserve the arrow of time information. No additional data augmentations are performed.

Table 1: Pretraining ablation experiments on CATER

| (a) Impact of the mask ratio | | (b) Impact of pretrain context | | (c) Impact of pretrain frames | |
|---|---|---|---|---|---|
| Mask ratio | Accuracy | # context | Accuracy | # frames | Accuracy |
| 37.5% | **69.48%** | 8 | **69.48%** | 32 | **69.48%** |
| 12.5% | 66.35% | 0 | 65.35% | 8 | 62.28% |
| 50% | 66.57% | 1 | 67.69% | 16 | 66.63% |
| 75% | 64.12% | 4 | 67.47% | 64 | 68.25% |
| 87.5% | 61.94% | 16 | 64.34% | | |

**Transfer learning:** For CATER, we evaluate on the static split which has 3,065 training, 768 validation, and 1645 test examples. We select the hyperparameters based on the validation performance, then use both training and validation data to train the model to be evaluated on the test split. By de-

Table 2: Ablation experiments on CATER tokens

| (a) Impact of the slot token number | |
| --- | --- |
| # slots | Accuracy |
| 1 | **69.48%** |
| 2 | 66.52% |
| 4 | 64.90% |
| 8 | 64.45% |

| (b) Impact of slot pool layer | |
| --- | --- |
| Pool layer | Accuracy |
| 11 | **69.48%** |
| 5 | 55.80% |
| 7 | 62.67% |
| 9 | 68.86% |

| (c) Impact of slot pool method | |
| --- | --- |
| Pool method | Accuracy |
| Slice | **69.48%** |
| Soft | 64.23% |
| Hard | 65.90% |

fault, we use 100 randomly sampled frames of 64×64 during training, and 100 uniformly sampled frames of stride 3 during evaluation. For ACRE, we explore all three splits, all of which contain 24,000 training, 8,000 validation, and 8,000 test examples. We again use the validation set to select hyperparameters and use both training and validation to obtain the models evaluated on the test split. We use all seven frames of 224×224 during training and evaluation.

**Default hyperparameters:** We use Adam optimizer for pretraining at learning rate of $10^{-3}$, and AdamW optimizer for transfer learning at learning rate of $5 \times 10^{-5}$. The pretraining checkpoints are trained from scratch for 1,000 epochs at batch size of 256. For transfer learning, we finetune the pretrained checkpoints for 500 epochs at batch size of 512. All experiments are performed on TPU with 32 cores. Below we study the impact of several key model hyperparameters.

## 4.2   ABLATION STUDY

We use CATER for ablation study in Table 1, and reuse the optimal hyperparameters in ACRE experiments. The impact of the number of slot tokens for ACRE is studied separately in Table 2.

**Masking ratio:** Contrary to the large masking ratio employed in vanilla MAE, we found that the optimal masking ratio was 37.5% on downstream CATER accuracy. This is perhaps due to the fact that CATER is designed to test "compositional generalization", and so the spatial context provides less information than in natural images and video.

**Number of Total Frames and Context Frames:** We also study the impact of the number of frames the implicit symbolic concept learner is pretrained on, and find the best performance on 32 frames. Fixing the total number of pretraining frames, we then ablate over the number of context frames, which are the frames from which slot representations are generated. When 0 context frames are used, we essentially utilize only patch-level representations to perform reconstruction with the temporal transformer (simulating a per-frame MAE followed by a temporal transformer). We find that the best performance is achieved with 8 context frames, which balances the number of slot representations with patch-level representations.

**Number of Slot Tokens:** Another useful ablation is on the impact of the number of slots used for CATER and ACRE. In CATER, we find that only 1 slot token per frame is enough to solve the reasoning task. We believe that this may be due to how the reasoning objective of CATER is designed: to successfully perform snitch localization, the model need only maintain an accurate prediction of where the snitch actually or potentially is, and can ignore more detailed representation of other objects in the scene. Under the hypothesis that the slot tokens represent symbols, perhaps the singular slot token is enough to contain the snitch location. On the other hand, when ablating over the number of tokens for the ACRE task (Table 3), we find that a higher number of tokens is beneficial for reasoning performance. This can potentially be explained by the need to model multiple objects across evidence frames in order to solve the final query; under our belief that slot tokens are encoding symbols, multiple may be needed in order to achieve the best final performance.

**Slot Pooling Layer and Method:** We ablate over which layer to pool over to generate the slot tokens. The patch tokens are discarded after the pooling layer, and only the slot tokens are further processed by the additional Transformer encoder layers. As expected, it is desirable to use all image encoder layers to process both slot and patch tokens. Additionally, we also study the impact of slot pooling method, and observe that adding additional single-headed soft attention and Gumbel-max attention are outperformed by simply using the slot tokens directly.

Table 3: Ablation on ACRE compositionality (comp), systematicity (sys), and I.I.D. (iid) splits.

| # slots | comp | sys | iid |
|---|---|---|---|
| 1 | 91.75% | 90.34% | 90.96% |
| 2 | 90.82% | 88.21% | 88.73% |
| 4 | 93.03% | **92.36%** | **92.13%** |
| 8 | **95.54%** | 86.18% | 88.97% |
| 64 | 90.45% | 80.07% | 90.82% |

Table 4: Benchmark results on CATER (static).

| Method | Object-centric | Object superv. | Top-1 Acc. | Top-5 Acc. |
|---|---|---|---|---|
| OPNet (Shamsian et al., 2020) | ✓ | ✓ | **74.8%** | - |
| Hopper (Zhou et al., 2021) | ✓ | ✓ | 73.2% | **93.8%** |
| ALOE base (Ding et al., 2021) | ✓ | ✗ | 60.5% | 84.5% |
| ALOE++ (Ding et al., 2021) | ✓ | ✗ | 74.0% | 90.4% |
| Random Init. | ✗ | ✗ | 3.3% | 18.0% |
| R3D LSTM | ✗ | ✗ | 60.2% | 81.8% |
| R3D + NL LSTM | ✗ | ✗ | 46.2% | 66.9% |
| MAE (He et al., 2022) | ✗ | ✗ | 27.1% | 47.8% |
| VideoMAE (Tong et al., 2022) | ✗ | ✗ | 63.7% | 82.8% |
| IS-CL (ours) | ✗ | ✗ | **69.5%** | **88.3%** |

## 4.3 COMPARISON TO THE STATE-OF-THE-ART

Table 4 compares the result of IS-CL against other state-of-the-art models on CATER snitch localization. We also compare IS-CL on ACRE against other existing models in Table 5. We pretrain MAE and VideoMAE ourselves on the same pretraining dataset and searched for their corresponding optimal hyperparameters. We observe that the spacetime ViViT used by VideoMAE leads to collapsed training, and modified it to use factorized encoder. Other results are cited from the published results. IS-CL achieves the best performance among the approaches that do not dependent on explicit object-centric representation, it also achieves overall state-of-the-art performance on the comp and iid splits of ACRE.

Table 5: Results on ACRE compositionality (comp), systematicity (sys), and I.I.D. (iid) splits.

| Method | Object-centric | Object superv. | comp | sys | iid |
|---|---|---|---|---|---|
| NS-OPT (Zhang et al., 2021) | ✓ | ✓ | 69.04% | 67.44% | 66.29% |
| ALOE (Ding et al., 2021) | ✓ | ✗ | **91.76%** | **93.90%** | - |
| Random Init. | ✗ | ✗ | 38.78% | 38.57% | 38.67% |
| CNN-BERT (Zhang et al., 2021) | ✗ | ✗ | 43.79% | 39.93% | 43.56% |
| MAE (He et al., 2022) | ✗ | ✗ | 80.27% | 76.32% | 80.81% |
| VideoMAE (Tong et al., 2022) | ✗ | ✗ | 78.85% | 71.69% | 77.14% |
| IS-CL (ours) | ✗ | ✗ | **93.03%** | **92.36%** | **92.13%** |

## 5 CONCLUSION AND FUTURE WORK

In this work we propose the implicit symbolic concept learner (IS-CL) framework, which trains a neural network end-to-end to solve complex visual reasoning tasks, without explicitly constructing an object-centric representation. IS-CL learns such implicit symbolic representations as slot embeddings in a pretraining step through a self-supervised video reconstruction objective via masking. We observe the exciting results that the learned representation behave like their symbolic counterparts, when measured on compositional generalization performance on CATER and ACRE benchmarks. Future work includes probing experiments to understand the information encoded by the slot tokens, and applying IS-CL to large-scale natural image and video datasets.

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
