# OpenReview forum: "Towards Learning Implicit Symbolic Representation for Visual Reasoning"
_ICLR.cc/2023/Conference — Submitted to ICLR 2023_

### Official Review · Reviewer_ejDU · 2022-10-21

**Confidence:** 4
**Clarity, Quality, Novelty And Reproducibility:** The paper is well presented and has n…
**Correctness:** 3
**Technical Novelty And Significance:** 3
**Empirical Novelty And Significance:** 3
**Recommendation:** 5

**Strength And Weaknesses:**

The paper presentation is clear and well-connected with existing works such as NS-CL and ALOE (Figure 1). There are a few places where the authors connect their work with Slot Attention, which has greatly helped me to understand the connections and differences between the proposed framework and others. The proposed framework is also relatively straightforward, with clear motivation. Experimental results have shown the success of the proposed framework. Furthermore, the ablation studies seem very adequate.

Talking about weaknesses, I will start with a few conceptual ones.

First, the idea of "concept learning," following the notations from Mao et al, primarily refers to the correspondence between linguistic units (words, phrases) and visual representations (e.g., red, cubic, left-of, etc.). This seems to be different from the definition of "concepts" or "implicit symbolic concept" in the paper, which, in my understanding, refers to "implicitly defined objects."

Second, when we talk about "visual concepts,", they are usually more "abstract" than "pixel reconstruction." However, since the overall training paradigm (pretraining time) is to reconstruct the pixels, I don't think there is evidence that the model is capable of discovering "symbolic concepts," e.g., colors, shapes, etc., from the pertaining. With all that, I am wondering if the title/model should be better phrased as "implicit object learning" or similar phrases?

Here're a few technical questions.

First, I am wondering why the authors have not tried visual question-answering benchmarks such as CLEVRER, especially given that based on the ablation study (Table 2a), CATER does not require an "implicit symbolic" representation (slot =1 then basically it's just a per-image representation). It seems that the framework can be easily extended to that, as in ALOE. And I think extending the framework to that can significantly improve the completeness of the paper.

Second, in Table 4, it seems that the model still underperforms several baselines. Although authors may argue that those better-performing algorithms are "object-centric" I think it is still completely reasonable to compare IS-CL with ALOE, because ALOE also does not use any object-detection labels during training. Again, I think adding a new benchmark that showcases the advantage of the proposed method will be ideal.

Third, more of a clarification question, regarding the comparison between VideoMAE and IS-CL. Is the only difference that you are using a "single-token" embedding for each frame whereas VideoMAE uses all visual tokens? Because these two models have very similar training objectives and similar architectures.

Fourth, is there any way that we can visualize the learned "implicit slot tokens?" For example, can you visualize the attention maps?

Finally, you mentioned that you have changed the slot-attention-style encoding with a customized transformer-style encoding. Do you have any ablation studies on that?

Clairification questions:

1. Page 5:  allowing the slots to attend not only to raw visual inputs, but also to the encoded patch-level representation. Can you be more specific about this?

**Summary Of The Paper:**

This paper proposes the framework "Implicit Symbolic Concept Learner" (IS-CL), a transformer-based architecture that is capable of reasoning about videos. The framework follows a pretraining-transfer-learning pipeline. That is, the model is first pretrained on a collection of unlabelled videos (by the MAE objective), and then finetuned on the target task (e.g., predicting the location of a certain object).

**Summary Of The Review:**

Overall I think this paper is a good contribution to the community. However, there are still a few issues that can be addressed by the authors before publication.

---

> ### Author Response · Authors · 2022-11-19
> **Our response**
>
> Thank you so much for your valuable insights and constructive review! We address your questions below:
>
> **Concept Learning**: We agree with the reviewer that concept learning is often referred to as correspondence between linguistic units and visual representations. However, we also believe that many of the intrinsic visual properties, such as objects, object parts, color, and shape, can be meaningfully grouped without the assistance of the language modality. We will clarify our wording and cite appropriate literature for clarity.
>
> **Implicit Object Learning**: Thank you for the suggestion! We would prefer “entities”, “symbols”, or “units”, over “objects”, as the latter implies a basic unit of representation is a single object. The IS-CL does not make the object-centric assumption, with the aim to learn a more flexible and general visual representation (of objects, object-parts, or spanning over multiple objects). We agree with the reviewer that learning the mapping from pixels to color or shapes likely involves cross-modal supervision.
>
> **Results on CLEVRER**: We fully agree and generalize our framework to support question answering on CLEVRER. The implementation details, and the preliminary results, can be found in the supplementary material (Section A.1, Table 6). We observe that CLEVRER achieves competitive performance on this benchmark. We plan to perform additional ablation studies on CLEVRER and include them in the final version.
>
> **Differences between VideoMAE and IS-CL**: Both methods are implemented with Video Transformers. During inference, the only difference is indeed the use of a few slot tokens (IS-CL) versus all image-patch tokens (VideoMAE). The main difference lies in the self-supervised pretraining stage: We propose to use the fully observed “context frames” which are encoded into “slots”, while VideoMAE applies random masking on all input frames and performs masked autoencoding. Finally, the VideoMAE implementation is based on a “SpaceTime” Video Transformer that performs self-attention among all spatiotemporal patches. We observed that this architecture led to training collapse and reimplemented VideoMAE with separate image encoders and a temporal transformer.
>
>
> **Understanding the learned representation**: We agree that visualizing the attention heatmaps would be very helpful to understand the learned representation. Due to the tight timeline, we took an alternative approach and quantified the behaviors of the learned representations via additional experiments in the supplementary material (Section A.2). Specifically, we try to measure the “locality” of information stored in the slot tokens. Our hypothesis is that the proposed IS-CL framework learns implicit symbolic representation which dynamically routes visual information into the encoded slot tokens of a transformer encoder. To validate our hypothesis, we perform a “probing” experiment testing the locality of information encoded in the slots. We first pretrain the IS-CL framework with K1 slot tokens. During finetuning, we remove a subset of the pretrained slot tokens and retain only K2 < K1 slot tokens. If the IS-CL pretraining leads to localized, implicit symbolic representations, then removing a subset of the tokens would lead to a significant drop in performance compared to the original model. Intuitively, it should also perform worse than a model pretrained with K2 tokens, which is pretrained to utilize a smaller set of tokens to encode implicit symbolic representation. In Table 7, we observe that when the slot tokens are removed, the resulting models are outperformed both by finetuning the original model (3rd row, right), and by models pretrained with the same smaller number of slots. The observations are in line with our hypothesis.
>
> **Relation with Slot Attention and Clarifying Page 5**: The goal of Slot Attention is to group pixels into object masks. As a result, the attention operation used by the Slot Attention algorithm always derives the keys and values from raw pixel inputs when computing the attention updates (Algorithm 1, line 8, in the Slot Attention paper). We generalize this formulation by iteratively updating the input representation from raw pixels (inputs to the first Transformer block) to visual representation encoded by the ViT (outputs of later Transformer blocks). We have clarified the relevant description on Page 5 to reflect your feedback.

---

> > ### Comment · Reviewer_ejDU · 2022-11-24
> > **Thank you for the clarification. More questions.**
> >
> > Dear authors,
> >
> > Thank you for your efforts during the rebuttal period. I think adding more clarifications about terminologies and comparisons with related works will definitely help.
> >
> > However, unfortunately, I am not fully convinced by your last two responses.
> >
> > 1. "Understanding the learned representation:" I think the comparison is not convincing enough. First, during such kind of "pretraining-finetuning" stages, many things can go wrong: e.g., hyperparameters, optimization strategies, etc. So the performance drop (which is not **significant**) can not be fully attributed to "having learned local representations." I think some more direct visualizations or downstream applications should be added to consolidate this point.
> >
> > 2. I am still confused about the relationship and comparison, especially conceptually, between this work and SlotAttention. I understand that they are different in terms of actual implementations: as mentioned by the authors, they use slightly different updating rules for the attention "slots." However, as far as I understand, the performance of the proposed model is worse than ALOE on both benchmarks. If this is the case, can authors use a few concise sentences describing the conceptual difference between slot attention and the proposed model, e.g., in terms of the task they can solve? the format of the representation they can learn? or the underlying principles? I believe this will be greatly helpful for both me and other readers.

---

> > > ### Author Response · Authors · 2022-11-29
> > > **Thank you for your questions! Follow up answers.**
> > >
> > > Dear reviewer ejDU,
> > >
> > > Thank you so much for your feedback! We would like to address your additional questions below:
> > >
> > > _1.1 Comparison (of missing slots) is not convincing enough_
> > >
> > > This is a fair concern, and we acknowledge that different hyper parameters may also lead to performance gaps. We would like to clarify that the performance drops are significant: Removing 3 out of 4 slots (by comparing the first row on the left, and the last row on the right) leads to 9.51% / 19.13% / 12.11% absolute accuracy drop on comp / sys / iid splits respectively. Removing 2 out of 4 slots (by comparing the second row on the left, and the last row on the right) leads to even bigger drops.
> > >
> > > _1.2 Direct visualizations_
> > >
> > > To further strengthen our point, we provide additional visualizations of the attention heatmaps from the slot tokens to the image patches. We first show the comparison of 4-slot and 1-slot models after pretraining: [Pretrain Attention Figure](https://i.imgur.com/PzAbAvb.png)
> > >
> > > We make two observations: (1) The attention heatmaps from both models learn to focus on objects; (2) Interestingly, the 4-slot model learns to focus on complementary objects, while the 1-slot model learns to focus on everything foreground. The observations confirm that object-centric representation emerges via our pretraining objective as encoded tokens.
> > >
> > > We then show the 4-slot model attention map after finetuning on the ACRE task: [Finetune ACRE Figure](https://i.imgur.com/ayyRdlM.png)
> > >
> > > We again observe that the heatmaps tend to focus on objects, which intuitively are important to solve the ACRE reasoning task. We also observe that most of the heatmaps focus on object relations (multiple objects, or object and the platform), which highlights that the encoded tokens can be "finetuned" and adjusted to the downstream tasks.
> > >
> > > _2.1 Differences between Slot Attention and IS-CL_
> > >
> > > Slot Attention focuses on object discovery tasks and has been mainly evaluated on object localization benchmarks. Building accurate object-based representation is a requirement. IS-CL focuses on learning _implicit symbolic representation_ that performs as well as their object-centric counterparts. They can be optionally _probed_ and visualized (as shown by the figures above) but that is neither the main goal nor a requirement. Conceptually, it would be arguably more interesting to discover representation alternative to object-centric representation that performs similarly or better on the challenging reasoning benchmarks. We would be happy to include this discussion in the final draft.
> > >
> > > _2.2 Slight different updating rules ... worse than ALOE on both benchmarks_
> > >
> > > We would like to clarify that as discussed in our general response, ALOE leverages an explicit object-centric representation and build that into a two-stage learning framework. Conceptually, the object detection can be performed by any algorithm (e.g. Slot Attention, MONET, or supervised Faster RCNN) but the object-centric assumption remains the same. IS-CL does not use this assumption. As a minor clarification, we do observe that IS-CL achieves on par or better performance than ALOE on ACRE.

---

> > > > ### Comment · Reviewer_ejDU · 2022-11-30
> > > > **Re: Thank you for your questions! Follow up answers.**
> > > >
> > > > Thanks for sharing the new results and the clarifications. So my understanding is:
> > > >
> > > > 1. ISCL actually learns object masks on the datasets studied by the authors.
> > > > 2. ISCL outperforms ALOE on ACRE but underperforms ALOE on CATER and CLEVRER (correct me if I am wrong).
> > > > 3. The main difference is that ALOE focuses on learning object-centric representations (i.e., each slot corresponds to one object), but IS-CL is more general, for example,  each slot can correspond to one property of one single object (e.g., color).
> > > >
> > > > Are these points correct?
> > > >
> > > > So I think I am still not fully convinced by the last point because this has not been illustrated with experiments thoroughly. As a reader of the paper, I am not seeing how ISCL is actually learning slots other than objects. While I understand that ISCL is a more general framework than ALOE (and of course, they have different architecture, training objectives, 2 stage vs 1 stage, etc.), I do not think the current paper shows its advantage.
> > > >
> > > > Given the interesting visualization results and the SOTA performance on ACRE, I will not object to accepting this paper. But I think it will be ideal to have experiments (even just a toy example) that better test the effectiveness of "implicit symbols".

---

> > > > > ### Author Response · Authors · 2022-11-30
> > > > > **Thank you for your feedback!**
> > > > >
> > > > > Dear reviewer ejDU,
> > > > >
> > > > > Yes, these points are correct, and we really appreciate your feedback!

---

### Official Review · Reviewer_LKo8 · 2022-10-24

**Confidence:** 4
**Correctness:** 2
**Technical Novelty And Significance:** 2
**Empirical Novelty And Significance:** 2
**Recommendation:** 5

**Clarity, Quality, Novelty And Reproducibility:**

The paper is well written with sufficient discussion on related works, even though the presentation of figures can be improved. The authors claim that code will be released upon acceptance of the manuscript.

**Strength And Weaknesses:**

Strength:

1. This paper proposes a powerful end-to-end network for compositional visual reasoning, and achieved nice results on two datasets that were regarded as (and designed to be) challenging for end-to-end systems.

2. The paper also demonstrates self-supervised pre-training can lead to useful representations for visual reasoning, which sheds light to further research.

3. The paper is well written and backed by rich experimental results.

Weakness

1. The paper claims that the model learns "implicit symbolic representation" without a clear definition and elaboration. What is the difference between "symbolic representation" vs. a regular latent representation learned by other self-supervised model? Moreover, there isn't sufficient study on the representation itself other than transfer learning results to back these claims. How does the same architecture perform without the representation?

2. The experimental results have not established, decisive evidence that end-to-end method is superior to object-centric representations. Specifically, under same supervision, the model does not significantly outperform ALOE and ALOE++. The paper also does not provide evaluation on the CLEVRER dataset, a video reasoning benchmark where object ALOE performs nicely.

3. Minor: The data flow in figures goes upwards instead of downwards, which might cause unnecessary confusions to readers.

**Summary Of The Paper:**

This paper proposes a method for compositional visual reasoning in videos, based on recent advances in self-supervised pretraining. The method first trains a spacial-temporal transformer that reconstructs the video frames under the masked autoencoder paradigm, then performs reasoning via transfer learning. The authors claim and demonstrate via extensive experimental study on the CATER and ACRE datasets, that a generalizable compact representation is learnt, superior to other approaches based on object-centric representation. Ablation studies are conducted.

**Summary Of The Review:**

This paper is an interesting paper that applies state-of-the-art self-supervised pretraining to compositional visual reasoning. Despite the reasonable results achieved on two challenging datasets, it is unfortunate that the paper does not provide a deeper look into the learned representation in order to back the claimed novelty. There is definitely a lot of room for improvement here. My score is borderline reject.

---

> ### Author Response · Authors · 2022-11-19
> **Our response**
>
> Thank you so much for your careful review and we really appreciate your constructive feedback! We address your questions below:
>
> **Define “implicit symbolic representation”**: It refers to vector-based representations from end-to-end trained neural networks that exhibit similar (compositional) generalization behaviors as their symbolic counterparts. Some of the learned implicit representations may be discretized into human-interpretable symbols at will, for the purposes of human understanding of and feedback to the model. Others may correspond to part, or a combination of human-interpretable symbols. As opposed to explicit symbolic representation (e.g. object detection), implicit symbolic representations do not require pre-defining a concept vocabulary or constructing concept classifiers. The implicit representation can also be “finetuned” directly on the target tasks and does not suffer from the early commitment or loss of information issues which may happen when visual inputs are converted into explicit symbols or frozen descriptors (e.g. via object detection and classification).
>
> **How does the same architecture perform without the representation**: We provide this ablation study in Table 4 (CATER) and Table 5 (ACRE). The “Random Init.” baseline uses the exact same architecture but without IS-CL pretraining. In addition, our reimplemented VideoMAE baseline uses a very similar architecture (patch tokens as opposed to slot tokens are fed into the temporal transformer) and is pretrained using their proposed objective. IS-CL outperforms both baselines significantly.
>
> **Performance gain over ALOE**: We see our on-par (and sometimes better) performance with ALOE as a strength, as our method does not require explicitly constructing object-level representations nor running object detectors, while ALOE and the neuro-symbolic approaches do.
>
> The goal of our paper is not to show end-to-end methods are superior to object-centric representations. Rather, the literature has well established the effectiveness of object-centric frameworks when the object-centric assumption holds. We see one of the main contributions of our work as demonstrating the effectiveness of “implicit symbolic representations” in end-to-end neural networks on challenging visual reasoning tasks. This line of research is not mutually exclusive, but complementary to approaches that use explicit symbolic representations, especially when the object-centric assumption holds true for the target tasks.
>
> **Results on CLEVRER**: Please kindly refer to our updated supplementary material. Our preliminary results confirm that IS-CL achieves comparable (and sometimes better) performance as ALOE on CLEVRER. We haven’t been able to perform extensive ablation study or hyperparameter search due to the tight timeline. They will be included in the final version.
>
> **Figure**: We have updated the figure caption based on your suggestion. We will try to update the figure itself to be clearer.
>
> **Deeper look into the learned representation**: We added more ablation studies in the supplementary material to understand the learned representation better. Specifically, our hypothesis is that the proposed IS-CL framework learns implicit symbolic representation which
> dynamically routes visual information into the encoded slot tokens of a transformer encoder. To validate our hypothesis, we perform a “probing” experiment testing the locality of information encoded in the slots. We first pretrain the IS-CL framework with K1 slot tokens. During finetuning, we remove a subset of the pretrained slot tokens and retain only K2 < K1 slot tokens. If the IS-CL pretraining leads to localized, implicit symbolic representations, then removing a subset of the tokens would lead to a significant drop in performance compared to the original model. Intuitively, it should also perform worse than a model pretrained with K2 tokens, which is pretrained to utilize a smaller set of tokens to encode implicit symbolic representation. In Table 7, we observe that when the slot tokens are removed, the resulting models are outperformed both by finetuning the original model (3rd row, right), and by models pretrained with the same smaller number of slots. The observations are in line with our hypothesis.

---

> > ### Author Response · Authors · 2022-11-30
> > **Follow up**
> >
> > Dear reviewer LKo8,
> >
> > Thank you again for your constructive feedback!
> > We are following up for your feedback on our clarifications on "implicit symbolic representation" / model architecture, and your requested additional results on CLEVRER and visualizations (linked in our response to reviewer ejDU). Thanks!

---

### Official Review · Reviewer_YXhg · 2022-10-30

**Confidence:** 2
**Correctness:** 3
**Technical Novelty And Significance:** 3
**Empirical Novelty And Significance:** 3
**Recommendation:** 6

**Clarity, Quality, Novelty And Reproducibility:**

Both the clarity and quality are good. The proposed framework may bring some fresh air to the community. The reproducibility is upon the code and pre-trained models released.

**Strength And Weaknesses:**

The studied direction is important, and the proposed method took a further step toward a general solution to visual reasoning. Figure 1 clearly demonstrates the difference and advantages between the proposed method and the previous approaches: the proposed framework can perform visual reasoning with neural networks and non-predefined implicit tokens. All the used components: slot token, MAE objective, and transformers (encoder, temporal) are reasonable. Overall the paper is easy to follow.

Extensive experiments are conducted in CATER and ACRE datasets and show the proposed framework consistently outperforms the previous approaches. Ablation studies are conducted thoroughly around masking ratio, total frame numbers, context frame numbers, slot token numbers, slot pool layer, and slot pool methods.

Many analyses and explanations in the current manuscript are intuitive and without supporting empirical evidence. For example, in "Number of Slot Tokens" section, experiments found 1 works the best in the CATER benchmark, and the author explained why: "the model need only maintain an accurate prediction of where the snitch actually or potentially is". If we add golden snitches up to 2/3/4 (if possible, or in other sims), will the best performance achieve by 2/3/4 implicit slot accordingly? or the best performance is still achieved by 1 slot.

What will the performance change if we set the slot token number to 100? will the proposed framework lose the "representational bottleneck" properties and lead to drastic performance drops?

In the transfer learning setting, the multi-class classification formulation for the goal of both CATER and ACRE tasks encodes strong human priors. The tested visual reasoning tasks are actually solved with specific designs for specific tasks in the end, while the studied implicit representations are uniform, and thus, the contribution did not weak a lot.

**Summary Of The Paper:**

This paper proposed an end-to-end implicit symbolic representation learning framework for visual reasoning tasks. It wisely adopts slot tokens for its bottleneck information properties, masked autoencoding objective, and transformers to learn implicit representations in a self-supervised way. The learned implicit representation can later be applied to solve specific visual reasoning tasks with proper head fine-tuning with target data. Such a learning framework has a very advantage: it does not require specific object abstraction (e.g., detection mask) and thus serves more general purposes. Results on two common benchmarks (CATER and ACRE) show consistent improvements achieved by the proposed framework.

**Summary Of The Review:**

In general, the reviewer believes the proposed approach takes a step towards visual reasoning with deep learning, while the analysis is intuitive and probably lacks some empirical analysis for strong support.

--------------
After discussing with AC and all other reviewers, we generally believe the current manuscript lacks some thorough analysis of the proposed framework, especially for 1 slot token works pretty well in many scenarios. I encourage the author to take all the points raised in the review in their revision and also keep improving the performance in various benchmarks. Reviewer ejDU lists the key points we discussed in our meeting, and hope them can help contribute to a better revision.

I appreciate the efforts made by the authors.

---

> ### Author Response · Authors · 2022-11-19
> **Our response**
>
> Thank you so much for your careful review and we really appreciate your constructive feedback! We address your questions below:
>
>
> **Adding more snitches**: This is a great suggestion! Unfortunately we do not have enough time to set up data generation and perform the necessary ablation studies to explore more snitches. However, according to our preliminary results on the CLEVRER dataset (available in supplementary), we also observed that using 4 slots is better than a single slot.
>
> **Using a much larger number of slots**: We perform the requested ablation and include the result in the updated Table 4. We observe a consistent drop in performance when using 64 slots compared to using 4 slots. However, the drop is much more significant on the sys split than on the other two splits. Further increasing the number of slots leads to an out-of-memory during pretraining.
>
> **CATER and ACRE formulations are limited**: We acknowledge this suggestion and have included preliminary results on CLEVRER in the supplementary materials.

---

> > ### Comment · Reviewer_YXhg · 2022-11-24
> > **Reviewer Response**
> >
> > I thank the efforts made by the authors. I encourage the author to improve their manuscript with all the feedback received in ICLR.
> >
> > The more snitches experiment is interesting. It should be included even if the results are bad, since the results can provide insights about the studied problem and the proposed framework.

---

> > > ### Author Response · Authors · 2022-11-30
> > > **Thank you for feedback!**
> > >
> > > Dear reviewer YXhg,
> > >
> > > We really appreciate your feedback and will incorporate feedback from all reviewers into the final version, thank you!

---

### Official Review · Reviewer_Cwxs · 2022-11-04

**Confidence:** 4
**Correctness:** 3
**Technical Novelty And Significance:** 2
**Empirical Novelty And Significance:** 2
**Recommendation:** 5

**Clarity, Quality, Novelty And Reproducibility:**

**Quality**: The paper has concrete motivation (the benefits of self-supervised approaches compared to stronger object-attribute annotations and the potential of temporal dynamics to encourage object discovery) and explores an important domain. While it would be good to experiments on more datasets, the ones presented are extensive, and explore multiple axes of comspositional generalization, performance on downstream reasoning tasks, and sensitivity to variations and ablations.

**Clarity**: The paper is well-written, the idea is clearly presented and it is accompanied with useful diagrams and visualizations.

**Novelty**:
The novelty of the idea is a bit limited, as there was a prior paper called ALOE (Attention over learned object embeddings enables complex visual reasoning) that, like the submitted paper, also used a self-supervised approach with a masked loss over frames and a transformer architecture, and it explores it over both on CATER and ACRE as done here as well as CLEVERER.

There is a difference between the approaches where ALOE uses representations extracted from Monet while the approach here uses ViT, but from prior experience with working on the CLEVR dataset in particular, I think prior models shown with attention they can in a quite straightforward manner manage to pinpoint specific objects from the raw 2d image representation, when coupled with the right losses and overall model and settings, so the new component in the paper doesn’t substantially contribute on addressing an unsolved technical challenge.

**ViT for object-centric on CLEVR vs. real-world**: This is especially straightforward given that the paper explores the model over the CLEVR dataset, since  they use ViT tokens to represent slots, and in CLEVR with choosing the right resolution, they could correspond approximately 1:1 to CLEVR objects – since they cover local convex regions, compared e.g. to more complicated objects or to non-object real-world regions such as sky etc.

**ViT vs Monet/sparser approaches**: It’s unclear to me whether ViT would the ideal approach in discovering objects on more general data, or since considering each slot on a dense 2D map to be an “object” leads to too many false-positive objects, and may struggle to find good correspondences on other datasets. This is a disadvantage compared to Monet used in ALOE, and there are also other object-centered models that discover sparser more compact representations such as Slot Attention, SAVI (extending slots attention to video and explores more diverse video datasets and also a bit of real-world robotic data), and the GroupViT model – which works very well on diverse real-world images.

**Claim correctness in introduction**: Furthermore, I find the conceptual distinction the paper tries to make in the introduction between ALOE which uses Monet vs this paper which uses ViT not compelling: the paper says the former is based on pre-processed extracted objects while the latter train the ViT together with the model, but Monet is an unsupervised approach too and one could think of training it together with the frame prediction explored in ALOE.

**Results**: The quantitative results of the paper are in line with prior works but not surpassing them and in particular not surpassing ALOE. Combined together with the novelty issue compared to ALOE that’s a main weakness of the paper.


**Strength And Weaknesses:**

**Strengths**:
* **Self-supervised**: The approach is self-supervised and so doesn’t require annotations like alternative approaches for building compositional scene representations such as object detectors etc.
* **dynamics for object-centered representations**: The approach encourages temporal dynamics to be captured in the representations, by using a self-supervised approach over videos. This is an important and still underexplored signal in the domain of self-supervised compositional learning that could greatly help discover objects.
* **Compositional Generalization**: The approach explores and show good results on compositional and systematic generalization, which gives good indication that the representations learned indeed disentangle the dimensions needed for the downstream reasoning tasks.
* **Extensive set of experiments on CLEVR-based data**: see quality point in the question below.

**Weaknesses**:
* **Experiments on synthetic datasets only**: The model is explored on CLEVR-based datasets only. While CLEVR-based data is definitely a great and suitable for exploring the presented approach, I would strongly encourage the authors to explore the approach either on new synthetic multi-object datasets with greater realism and diversity.


**Summary Of The Paper:**

A self-supervised training over video is proposed for learning compositional scene representations. The paper explores the approach on the CATER and ACRE datasets (video versions of CLEVR), and show its benefits in the domain of visual reasoning and question answering over these videos.

**Summary Of The Review:**

The paper shows a new approach in a growing relatively under-explored domain. Multiple experiments are presented on two datasets, the paper is easy to follow and has good visualizations, but the main issues are 1) the bit limited novelty of the idea in light of the ALOE paper that explored self-supervised transformer based approaches for both the dataset explored here and an additional one (see details above), and 2) The fact that the paper doesn’t explore other datasets beyond the CLEVR family. I therefore overall recommend weak rejection at this point and recommend exploring how to extend the approach for additional datasets, which may lead also to new modeling ideas that could strengthen the technical contribution of the paper.

---

> ### Author Response · Authors · 2022-11-19
> **Our response**
>
> Thank you so much for your careful review and constructive comments!
>
> **Experiments on synthetic datasets only**: We acknowledge this limitation. In the updated supplementary material, we added experimental results on the CLEVRER dataset, which are in line with our observations on CATER and ACRE. We believe exploring newer synthetic datasets with greater visual complexity is a great suggestion, we also plan to explore diagnostic real video benchmarks (e.g. the “Perception Test” recently released by DeepMind) as a future direction.
>
> **Novelty to ALOE**: Please kindly refer to our general response. In addition, a key difference between object-centric representation learned by MONet and our representation encoded by ViT is that the former is fixed and “frozen”, while the latter is updated via finetuning. In Table 4 and 5, we show that attending to objects is not straightforward even on the synthetic data: A randomly initialized baseline with the same neural architecture achieves very poor performance on both tasks.
>
> **Is ViT suitable for discovering objects**: As the reviewer kindly suggested, there is evidence supporting that ViT can discover objects with proper (supervised / self-supervised / cross-modal) objectives, such as GroupVIT, or DINO. We believe that testing our approach on real-world benchmarks is an exciting direction to explore.
>
> **MONet can be trained together with ALOE**: This is a great suggestion, and our paper indeed thrives to provide an answer to this question (namely, can we learn an end-to-end neural network directly from pixels to solve visual reasoning tasks?), which was not discussed in the original ALOE paper. Furthermore, we also thrive to answer whether the object-centric assumption needs to be provided explicitly.
>
> **Performance**: As discussed in our overall response, we see one of the main contributions of our work as demonstrating the effectiveness of “implicit symbolic representations” in end-to-end neural networks on challenging visual reasoning tasks. This line of research is not mutually exclusive, but complementary to approaches that use explicit symbolic representations, especially when the object-centric assumption holds true for the target tasks.

---

> > ### Author Response · Authors · 2022-11-30
> > **Follow up**
> >
> > Dear reviewer Cwxs,
> >
> > Thank you again for your constructive feedback!
> > We are following up for your feedback on our response above, the updated manuscript / supplementary materials, as well as additional results on CLEVRER and visualizations (linked in our response to reviewer ejDU). Thanks!

---

> > ### Comment · Reviewer_Cwxs · 2022-12-07
> > **Response to authors**
> >
> > Dear authors,
> > Thank you very much for your response and efforts!
> > I think given the current state of the paper I'd like to keep the score I gave since it best captures my estimate but I do warmly recommend the authors to pursue the suggestions for improvements, extensions and further exploration to bolster the paper.
> > Best of luck!
> > - Reviewer

---

### Author Response · Authors · 2022-11-19
**Overall response**

We would like to thank all reviewers for your thoughtful reviews and constructive feedback! We are glad that reviewers found our evaluations extensive (Cwxs, LKo8), our proposed method explores important and underexplored signal (Cwxs), and our paper is well presented (ejDU) and brings fresh air to the community (YXhg). We highlight our responses to the important questions brought by the reviewers below:

**Implicit symbolic representation**: refers to vector-based representations from end-to-end trained neural networks that exhibit similar (compositional) generalization behaviors as their symbolic counterparts. Some of the learned implicit representations may be discretized into human-interpretable symbols at will, for the purposes of human understanding of and feedback to the model. Others may correspond to part, or a combination of human-interpretable symbols. As opposed to explicit symbolic representation (e.g. object detection), implicit symbolic representations do not require pre-defining a concept vocabulary or constructing concept classifiers. The implicit representation can also be “finetuned” directly on the target tasks and does not suffer from the early commitment or loss of information issues which may happen when visual inputs are converted into explicit symbols or frozen descriptors (e.g. via object detection and classification).

**Comparison with ALOE**: We see the explicit object-centric representation used by ALOE as the key difference from our work. Conceptually, ALOE needs to explicitly construct an input representation based on the object-centric assumption. The object-centric assumption is fixed and the representation is not updated for the downstream tasks. IS-CL learns to construct implicit symbolic representation without using the object-centric assumption (Reviewer YXhg: IS-CL “does not require specific object abstraction”), and the learned representations (encoded slot tokens) can be adjusted for different downstream tasks via backpropagation. Implementation-wise, our framework can be implemented with minor modifications to the general transformer encoder network, while ALOE requires an additional (unsupervised) object detector (MONet). Performance-wise, we see our on-par (and sometimes better) performance with ALOE as a strength, as our method does not require explicitly constructing object-level representations nor running object detectors, while ALOE and the neuro-symbolic approaches do. Finally, we see one of the main contributions of our work as demonstrating the effectiveness of “implicit symbolic representations” in end-to-end neural networks on challenging visual reasoning tasks. This line of research is not mutually exclusive, but complementary to approaches that use explicit symbolic representations, especially when the object-centric assumption holds true for the target tasks.

**CLEVRER results**: We have included results on the CLEVRER benchmark as requested (available in the supplementary), which confirms that IS-CL achieves comparable (and sometimes better) results as ALOE on CLEVRER. We haven’t been able to perform extensive ablation study or hyperparameter search due to the tight timeline. They will be included in the final version.

---

### Decision · Program_Chairs · 2023-01-20

**Decision:**

Reject

**Justification For Why Not Higher Score:**

The paper shares some technical similarities with prior work ALOE, does not uniformly improve upon ALOE in quantitative performance, and does not convincingly argue that the "implicit symbolic" representations learned by the method are different from object-centric representations used in prior work.

**Justification For Why Not Lower Score:**

N/A

**Metareview: Summary, Strengths And Weaknesses:**

The paper presents a method for learning compositional scene representations from unlabeled video using an approach inspired by masked autoencoders (MAE). The resulting representations can then be fine-tuned for downstream visual reasoning tasks. The paper presents extensive experiments on the CATER and ACRE datasets, and is generally well-written. Some of the initial issues raised by reviewers were resolved by the author responses, including results on more datasets (authors added results on CLEVERER), several additional ablations (suggested by Reviewer YXhg), more experiments on what the learned representations capture (experiments added to supplementary), and a more precise meaning of “implicit symbolic representations.”

After the author's responses, the reviewers gave borderline ratings to the paper (with ejDU leaning slightly toward acceptance, Cwxs and LKo8 slightly favoring rejection, and only YXhg in favor of acceptance). In order to come to a decision on this paper, the AC and reviewers met to discuss it.

From the discussion, it became clear that there are several key issues with the paper that had not been fully addressed in the author’s responses. First is the performance relative to ALOE. Both the proposed method (IS-CL) and ALOE receive the same supervision: unlabeled videos and input/output pairs for downstream tasks, and no ground-truth object locations, program execution traces, or other intermediate supervision. However IS-CL demonstrates mixed performance relative to ALOE, outperforming ALOE on ACRE but underperforming it on CATER and CLEVERER. Second is the distinction between the “implicit symbols” that IS-CL claims to learn, and the object-centric representation used by ALOE; the authors argue that these are distinct, but were not able to convincingly demonstrate that IS-CL actually learns non-object-centric representations; indeed in the discussion with Reviewer ejDU it became clear that IS-CL primarily learns to attend to objects. Third, the datasets used could have been more convincing; in their initial review, Reviewer Cwxs suggested showing results on more realistic, non-CLEVR-like datasets, and felt that this would make the results much more convincing; fpr example, comparing to the experimental setups of e.g. SAVi, SAVi++, or GroupVIT would have made the paper much stronger.

While Reviewers ejDU and YXhg gave positive ratings to the paper following the author feedback, in the discussion both agreed that the above were critical weaknesses, and neither was willing to argue that the paper should be accepted. In the end all reviewers agreed that the paper is not ready for publication in its current form.


**Summary Of Ac-Reviewer Meeting:**

We discussed several issues about the paper:
(1) Comparison relative to ALOE. The proposed method does not uniformly outperform ALOE. However reviewers also felt that even if the paper had been written exactly the same but had outperformed ALOE on all tasks, it still would have had weaknesses. The authors argue that the two are not comparable since ALOE is "object-centric", but reviewers found this unconvincing since they both receive the same supervision, and also due to (2).
(2) Implicit symbolic vs object-centric representations. The authors argue that these are distinct, but were not able to convincingly demonstrate that IS-CL actually learns non-object-centric representations; indeed in the discussion with Reviewer ejDU it became clear that IS-CL primarily learns to attend to objects.
(3) Datasets. Reviewer Cwxs suggested showing results on more realistic, non-CLEVR-like datasets, and felt that this would make the results much more convincing; fpr example, comparing to the experimental setups of e.g. SAVi, SAVi++, or GroupVIT.
(4) Simplicity vs ALOE. The AC felt that despite these weaknesses, one benefit of the proposed method that had gone undiscussed was the  relative simplicity of the proposed method vs ALOE (including the unsupervised pretraining of the MONet object detector upon which ALOE relies). Reviewers saw this benefit, but this did not seem like a strong enough reason to counterbalance the above weaknesses.